# Comparison of commercially available, rapid, point-of-care C-reactive protein assays among children with febrile illness in southwestern Uganda

**Caitlin A. Cassidy**[1], **Lydiah Kabugho**[2], **Georget Kibaba**[2], **Bradley Lin**[3], **Brandon Hollingsworth**[4], **Emmanuel Baguma**[2], **Jonathan J. Juliano**[1,5], **Edgar M. Mulogo**[2], **Ross M. Boyce**[1,5,6], **Emily J. Ciccone**[5]*

1 Department of Epidemiology, Gillings School of Global Public Health, University of North Carolina at Chapel Hill, Chapel Hill, North Carolina, United States of America, 2 Department of Community Health, Mbarara University of Science and Technology, Mbarara, Uganda, 3 Washington University in St. Louis, St. Louis, Missouri, United States of America, 4 Department of Entomology, Cornell University, Ithaca, New York, United States of America, 5 Division of Infectious Diseases, School of Medicine, University of North Carolina at Chapel Hill, Chapel Hill, North Carolina, United States of America, 6 Carolina Population Center, University of North Carolina at Chapel Hill, Chapel Hill, North Carolina, United States of America

* emily_ciccone@med.unc.edu

## Abstract

In Uganda, children with febrile illness are often treated with antibiotics even though most have self-limiting, likely viral, infections. C-reactive protein (CRP) measurement can help identify those who are more likely to have a bacterial infection and therefore need antibiotic treatment. Implementation of a CRP rapid diagnostic test (RDT) at the point-of-care in resource-constrained settings with minimal laboratory infrastructure could reduce unnecessary antibiotic use. In this study, we evaluated the performance of three semi-quantitative CRP RDTs (Actim, BTNX, Duo) against a reference CRP assay requiring an electrically powered analyzer (Afinion). While both tests demonstrated substantial agreement with Afinion, Actim had slightly higher agreement than BTNX. The sensitivity was higher for the BTNX test, whereas the Actim test had a higher specificity, at cut-offs of 40 mg/L and 80 mg/L. At a cut-off of 20 mg/L, Duo demonstrated substantial agreement with the Afinion test as well. Our results demonstrate the reliability of CRP RDTs when compared to a reference standard. CRP RDTs without the need for a laboratory-based analyzer are promising tools for optimizing antibiotic use in low-resource settings.

## Introduction

The spread of antimicrobial resistance (AMR) is a global crisis that represents a growing threat to human health [1]. Often perceived as a problem of high-income countries, AMR is increasingly being recognized in resource-limited settings due to wider access to care and greater availability of broad-spectrum antibiotics [1–4].

A major driver of AMR is the misuse of antibiotics, including overuse for viral or non-infectious conditions [5]. In sub-Saharan Africa (SSA), acute febrile illness (AFI) is a common

**Data Availability Statement:** Deidentified individual data that supports the results will be shared following publication provided the

investigator who proposes to use the data has approval from an Institutional Review Board (IRB), Independent Ethics Committee (IEC), or Research Ethics Board (REB), as applicable, and executes a data use/sharing agreement with UNC. Researchers may apply for data access by contacting the corresponding author or the UNC IRB at irb_questions@unc.edu.

**Funding:** This study was funded by an Early Career Award from the Thrasher Research Fund to EJC (#15206). In addition, EJC was supported by the National Heart, Lung, and Blood Institute through Grant Award Number [5T32HL007106] during the study period, RMB is currently supported by the National Institute of Allergy and Infectious Diseases through Grant Award Number [5K23AI141764], and JJJ is currently supported by the National Institute of Allergy and Infectious Diseases through Grant Award Number [5K24AI134990]. Medilink Lab & Surgicals Limited (Kampala, Uganda) donated the Malaria/CRP Duo tests. The use of REDCap was supported by the National Center for Advancing Translational Sciences (NCATS), National Institutes of Health, through grant award number UL1TR001111. The content is solely the responsibility of the authors and does not necessarily represent the official views of the NIH. The funders had no role in study design, data collection and analysis, decision to publish, or preparation of the manuscript.

**Competing interests:** The authors have declared that no competing interests exist.

reason for pediatric care-seeking and antibiotic dispensation [6], although ample data suggest that a large proportion of febrile episodes are caused by self-limited viral infections that do not require antibiotic treatment [7–9]. This practice is in part because a small proportion of children with AFI will have bacterial infections, which may be life-threatening if not treated in a timely and appropriate manner; the fear of poor outcomes influences prescriber practices and favors over-treatment [10]. In addition, differentiating viral from bacterial infections in resource-limited settings is challenging, as diagnostic imaging and robust laboratory infrastructure are not widely available. Reliance on syndromic diagnosis contributes to both antibiotic overuse and an under-recognition of children at high-risk for bacterial infection [11, 12]. As such, the development and implementation of effective and low-cost rapid diagnostic tools to guide prescribing practices is imperative, as highlighted by the World Health Organization's strategy for optimizing antimicrobial use in its Global Action Plan on Antimicrobial Resistance [5].

C-reactive protein (CRP) is a protein produced by the liver and released as an acute phase reactant in response to inflammation. Importantly, CRP production is greater in response to bacterial as compared to viral infection [13]. Its measurement has been shown to help distinguish viral from bacterial infections using thresholds ranging from >10 mg/L to >125 mg/L [14–19]. Furthermore, studies have demonstrated that incorporation of CRP point-of-care testing into the evaluation of children presenting with febrile illness can significantly reduce antibiotic use without increasing adverse outcomes [20–23]. Historically, most commercially available CRP tests are costly or require an analyzer, limiting their use in many low-and-middle-income country (LMIC) settings, but several, more inexpensive, point-of-care rapid diagnostic tests (RDTs) that can be performed using capillary blood samples outside of health facilities are now available [24, 25].

Employing CRP RDTs in the diagnosis of febrile children in rural, resource-limited settings may aid in identifying those at high risk for bacterial infection requiring antibiotics, while also reducing overuse of antibiotics in low-risk children and expanding equitable access to diagnostic tools [26, 27]. However, additional data on the performance of these RDTs are needed prior to their routine implementation. Therefore, we enrolled young children presenting to an outpatient clinic in western Uganda with febrile illness and compared and evaluated the performance of three CRP RDTs against a more traditional, laboratory-based CRP assay requiring an analyzer.

## Materials and methods

### Ethics statement

Ethical approval for the study was obtained from the University of North Carolina Institutional Review Board (#18–2803), the Mbarara University of Science and Technology Research Ethics Committee (14/03-19), and the Uganda National Council for Science and Technology (HS 2631).

### Study setting and design

This prospective study took place in Bugoye sub-county, a rural, mountainous area in the Kasese District of western Uganda. The region experiences year-round malaria transmission and semiannual peaks after the two rainy seasons from September through December and March through May. We enrolled children who presented to the Outpatient Department (OPD) of Bugoye Health Centre III (BHC), a primary health center that serves a population of approximately 50,000 residents. There is a small laboratory onsite with trained staff who perform basic testing, including light microscopy, hemoglobin estimation, and RDTs.

## Study procedures

Upon presentation to the OPD, children were evaluated by a health center clinician who, together with study staff, identified children who were potentially eligible for the study. Inclusion criteria were (1) age 6 months to 5 years, and (2) either a fever (i.e., axillary temperature $\geq$ 38˚C) in clinic or a history of fever within the last 7 days. Study staff approached the parent or guardian of eligible children, and if written informed consent was provided, the child was enrolled. If the guardian did not provide consent, the child was evaluated and treated per local standards of care.

The participant's clinical presentation and medical history were recorded on a case report form (S1 Text). Study staff collected vital signs, including axillary temperature, respiratory rate, heart rate, oxygen saturation, and weight. Each participant was tested using a histidine rich protein-2 (HRP2)-based rapid antigen test (mRDT) for *Plasmodium falciparum* malaria as part of routine standard of care evaluation for fever [28, 29]. Study testing consisted of two rapid CRP tests (Actim CRP, Medix Biochemica, Norway; BTNX CRP; BTNX Inc., Canada) and a CRP test requiring an analyzer and electricity (Afinion CRP, Abbott Laboratories, United States). All tests were performed using capillary blood samples collected by finger prick. The first 50 children enrolled also underwent testing using a combination malaria and CRP test (STANDARD Q Malaria/CRP Duo, SD Biosensor, Republic of Korea). These CRP tests were selected because of their feasibility for implementation in a resource-limited facility, lower costs, and because they are CE marked. After study testing was completed, the child and caregiver then returned to the clinician for further assessment. CRP results were not provided to the clinicians as they were done for research purposes only.

## Laboratory testing

The Afinion, Actim, BTNX, and Malaria/CRP Duo tests were performed according to manufacturer instructions by one of three laboratory technicians in the clinical lab at BHC who were trained on use of each test. All tests for an individual participant were conducted by the same technician. Test characteristics are detailed in Table 1. The Actim and BTNX tests results are semi-quantitative, whereas the Duo test is a combination malaria rapid diagnostic test (positive/negative) and semi-quantitative CRP test. The Afinion CRP test run on the AS100 Analyzer is quantitative (range = 5–200 mg/L) and served as the reference standard. It requires the use of an electrically powered analyzer so cannot be conducted outside of facility-based clinical settings. It has previously been shown to have high correlation with conventional laboratory-based methods of CRP measurement [30–32]. The technicians were not blinded to the results of the reference test.

## Data analysis

Demographic and clinical characteristics were summarized using standard statistical measures. Nutritional status was assessed using calculated weight-for-age z-scores [36]. We calculated summary statistics of CRP levels by mRDT status and weighted Kappa statistics with corresponding 95% CIs to measure the agreement of the Actim and BTNX tests with the Afinion test, where the Afinion was categorized to align with the semi-quantitative categories of the RDTs [37]. We also calculated Fleiss' Kappa statistic with a corresponding 95% CI to measure the agreement between all three tests simultaneously [38]. In addition, we conducted McNemar's test for marginal probabilities between Afinion and both RDTs, using 40 mg/L as the cut-off for a positive test. The sensitivity and specificity, with corresponding 95% CIs based on the modification of the Wilson interval developed by Yu et al. [39], of the Actim and BTNX tests were calculated using Afinion as the reference test, for both 40 mg/L and 80 mg/L cut-offs for a positive test based on previous

**Table 1. Description of study C-reactive protein assays.**

| Test Name (Manufacturer) | Type of Test | Cut-offs or Analytical measuring interval (mg/L) | Sample Type | Time to Result | Previous Studies of Test Performance |
|---|---|---|---|---|---|
| Point-of-care Assays | | | | | |
| Actim® CRP (Medix Biochemica) | Semi-quantitative, immunochromatographic | <10, 10–40, 40–80, >80 | Whole blood | 5 min | [24, 30, 33] |
| Rapid Response™ CRP Semi-Quantitative Test Strip (Quad Line) (BTNX) | Semi-quantitative, immunochromographic | <10, 10–40, 40–80, >80 | Whole blood, serum, plasma | 5 min | [24]. Per package insert, >99.9% sensitivity and 97.5% specificity for any positive result compared to a leading commercial EIA test. |
| STANDARD™ Q Malaria/CRP Duo (SD Biosensor) | Malaria: qualitative, P.f/Pan Ag* CRP: Semi-quantitative, immunochromographic | >20 | Whole blood | 15 min | No published reports found. Clinical trial in process [34]. |
| Reference Assay | | | | | |
| Afinion™ CRP (Abbott Laboratories) | Quantitative, solid phase immunochemical | 5–200 (whole blood), 5–160 (serum, plasma) | Whole blood, serum, plasma | 4 min | [24, 30–32, 35] |

*P.f/Pan Ag: *Plasmodium falciparum* (HRP-2), and Plasmodium sp. (pLDH) antigen

studies of CRP measurement in the context of acute respiratory illness [14, 21, 22, 40, 41]. Global Wald tests were conducted to determine simultaneously whether the sensitivities and specificities of the Actim and BTNX RDTs differed from each other [41]. For the Duo CRP RDT only, we calculated a simple Kappa statistic with a corresponding 95% CI, where the Afinion was categorized to align with the binary results of the Duo CRP test. We also calculated a simple Kappa statistic with a corresponding 95% CI to compare the primary mRDT and the Duo malaria RDT. The sensitivity and specificity of the Duo test, with corresponding 95% CIs, were calculated using Afinion as the reference test [41]. We included only nonmissing test results for any given comparison. Analyses were conducted using SAS Studio 3.8 and R 4.0.2.

## Results

### Study population

From October to December 2020, we enrolled 150 children aged 6 months to 5 years who presented with fever to BHC. Demographic characteristics and details of the clinical presentation are shown in Table 2. In brief, 85 (57%) were male, the median age was 24 months (Q1-Q3: 13–36), and 11% (16/150) were severely underweight as defined by a weight-for-age of less than 3 standard deviations (SD) below the median. The most common presenting symptoms were fever, cough, rhinorrhea, and diarrhea; no children reported sore throat, bleeding, coma, or chest in-drawing. 59% (88/150) had a measured temperature of 38°C or greater at presentation to the OPD. The median duration of fever was 3 days (Q1-Q3: 2–7) at the time of presentation. Approximately 19% (29/150) reported having previously been seen for the same illness (e.g., at a drug shop or by a community health worker), and 15% (23/150) reported receiving either an antimalarial or antibiotic treatment in the prior 14 days.

### CRP testing

All individuals underwent CRP testing; results for all three CRP tests (Actim, BTNX, and Afinion) were available for 89% (134/150) of participants. The Afinion test was not able to be run

**Table 2. Demographic characteristics of the study participants (n = 150).**

| Demographic Characteristics | n (%) |
|---|---|
| *Age in months (median (Q1-Q3))* | 24 (13–36) |
| *Sex* | |
| Female | 65 (43) |
| Male | 85 (57) |
| *Parish* | |
| Bugoye | 63 (42) |
| Mubuku | 12 (8) |
| Muhambo | 26 (17) |
| Other | 49 (33) |
| **Clinical Characteristics** | |
| *Weight-for-age*[a] | |
| Severely underweight | 16 (11) |
| Moderately underweight | 23 (15) |
| Normal | 108 (72) |
| Overweight | 2 (1) |
| *Symptoms Reported by Caregiver* | |
| Fever | 144 (96) |
| Cough | 104 (69) |
| Runny nose | 50 (33) |
| Diarrhea | 37 (25) |
| Skin rash | 31 (21) |
| Not eating | 28 (19) |
| Headache | 25 (17) |
| Vomiting | 21 (14) |
| Abdominal pain | 8 (5) |
| Joint pains | 4 (3) |
| Muscle aches | 3 (2) |
| Fatigue | 3 (2) |
| Convulsions | 2 (1) |
| Fast breathing | 1 (1) |
| **Vital Signs** | |
| *Tachycardic*[b] | 12 (8) |
| *Tachypneic*[c] | 39 (27) |
| *Hypoxic*[d] | 17 (11) |

[a] Nutritional status categories based on the WHO Child Growth Standards 2006 [42]. Severely underweight (weight-for-age $<-3$ SD of the median), moderately underweight (weight-for-age $<-2$ SD and $\geq-3$ SD of the median, normal (weight-for-age $\geq-2$ SD and $\leq2$ SD of the median), overweight (weight-for-age $>2$ SD and $\leq3$ SD of the median). Weight missing for 1 participant.

[b] Tachycardia defined as $>160$ beats per minute (bpm) for age 2 months– 1 year, $>150$ bpm for ages 1–3 years, and $>140$ bpm for ages 3–5 years [43].

[c] Tachypnea defined as $\geq50$ breaths per min for ages 2 months to 1 year and $\geq40$ breaths per minute for ages 1–5 years [29]. Respiratory rate was not measured for 5 participants.

[d] Hypoxia defined as $<90\%$ oxygen saturation.

**Table 3. Performance of Actim and BTNX RDTs at cut-offs of 40 and 80 mg/L using the Afinion as the reference standard.** PPV: positive predictive value, NPV: negative predictive value.

| Estimate | Cut-off of 40 mg/L | Cut-off of 80 mg/L |
|---|---|---|
| | % (95% CI) | % (95% CI) |
| Prevalence of positive CRP | 26.9 (20.0, 35.1) | 11.5 (7.0, 18.1) |
| Actim | | |
| Sensitivity | 80.0 (64.3, 90.2) | 80.0 (55.3, 93.4) |
| Specificity | 90.5 (83.1, 95.1) | 90.4 (83.8, 94.7) |
| PPV | 75.7 (60.1, 86.8) | 52.2 (33.0, 70.8) |
| NPV | 92.5 (85.4, 96.4) | 97.2 (92.2, 99.2) |
| BTNX | | |
| Sensitivity | 94.3 (81.7, 98.7) | 86.7 (62.7, 96.8) |
| Specificity | 78.9 (69.8, 86.0) | 86.1 (78.7, 91.3) |
| PPV | 62.3 (48.9, 74.1) | 44.8 (28.4, 62.4) |
| NPV | 97.4 (91.2, 99.5) | 98.0 (93.2, 99.6) |

for 11% (16/150) individuals because of power interruptions, and the Actim test was not run for 3% (4/150) of participants because of supply chain delays. The performance of each of the point-of-care tests, Actim and BTNX, compared to Afinion using two different binary CRP cut-offs is shown in Table 3 and Fig 1. The sensitivity was higher for the BTNX test, whereas the Actim test had a higher specificity at both cut-offs. At a cut-off of 40 mg/L, the overall Wald test demonstrated statistically significant differences in the sensitivity and specificity of

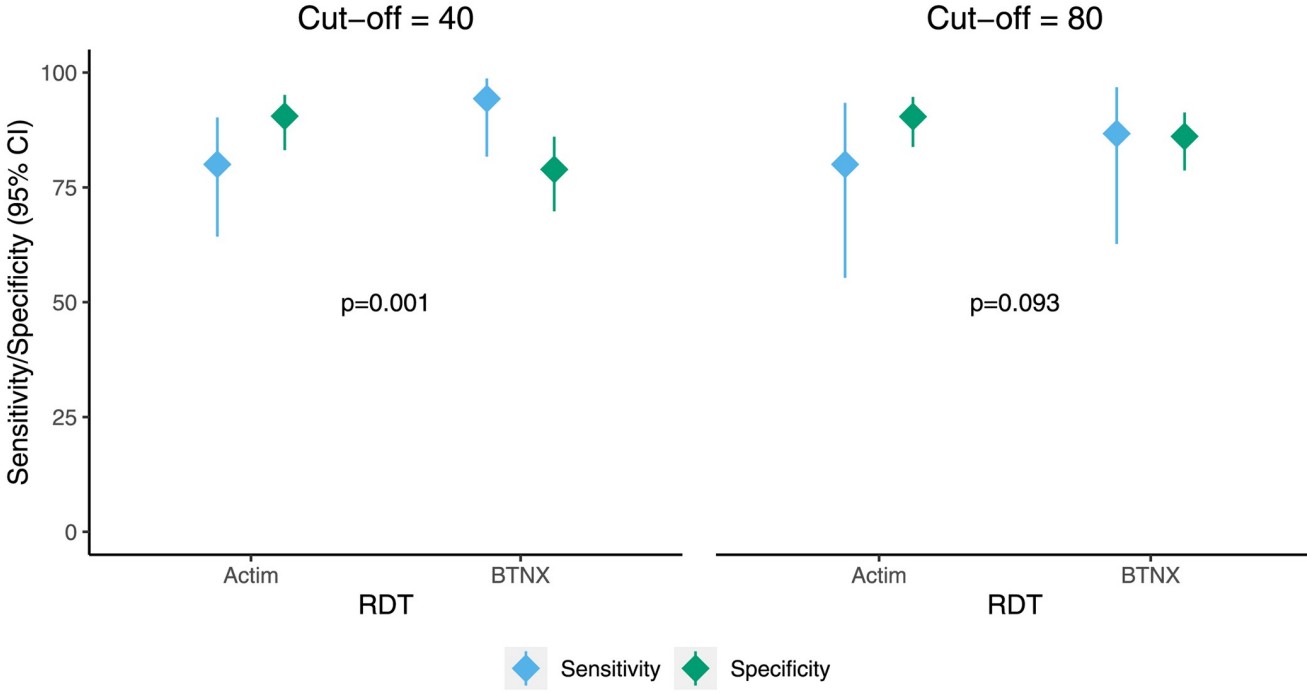

**Fig 1. Sensitivity and specificity of Actim and BTNX RDTs.** Sensitivity and specificity (diamonds) and corresponding 95% confidence intervals (lines) of Actim and BTNX RDTs at cut-offs of 40 and 80 mg/L with Afinion as the reference standard. P-values are for overall Wald test assessing whether sensitivities and specificities differ between Actim and BTNX.

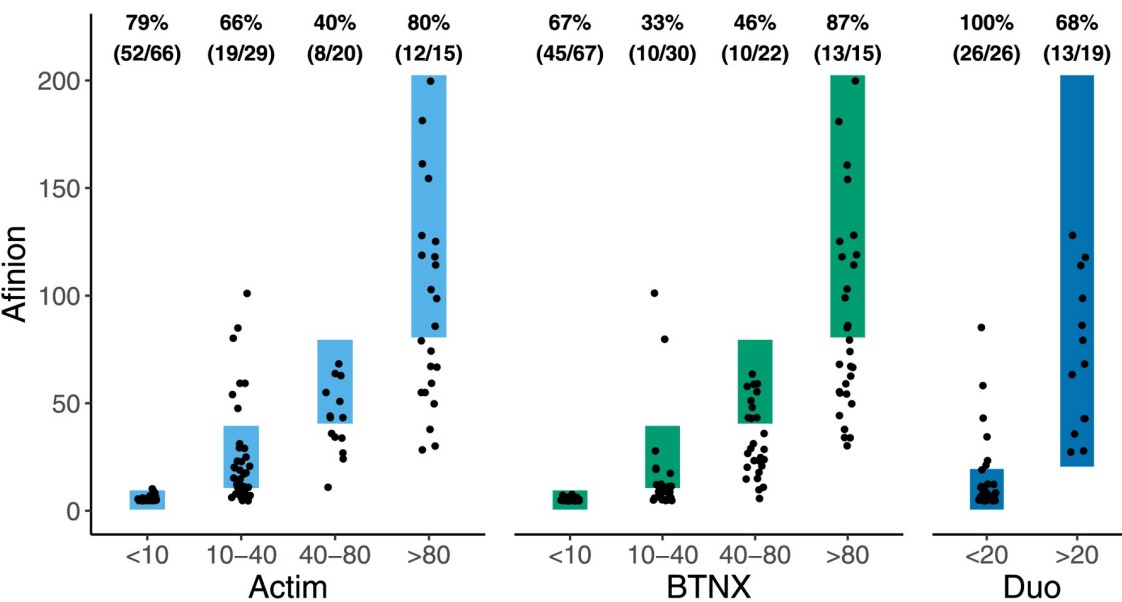

**Fig 2. Agreement between RDTs with Afinion.** Measured in 130 (Actim), 134 (BTNX), and 50 (Duo) participants. Points inside rectangles indicate perfect agreement between tests. Percentages represent the number of samples classified correctly by the RDT (numerator) divided by the number of samples in that category as classified by Afinion (denominator).

the Actim and BTNX tests (p = 0.001). At a cut-off of 80 mg/L, the sensitivity and specificity did not differ (p = 0.093).

As the output of the BTNX and Actim tests is semi-quantitative with four possible results, we calculated weighted and Fleiss' Kappa statistics to determine the agreement between the semi-quantitative tests and the reference test. The Actim test had slightly better agreement (weighted Kappa = 0.70 (95% CI 0.62 to 0.78) than the BTNX test (weighted Kappa = 0.62 (95% CI 0.54 to 0.70)) with the Afinion test. The agreement of CRP levels in the reference (Afinion) test compared to the Actim and BTNX tests is shown in Fig 2. Both semi-quantitative tests demonstrated less misclassification in the lower and upper categories (<10 and >80 mg/L) than the middle categories (10–40 and 40–80 mg/L). The three tests demonstrated lower agreement when compared simultaneously (Fleiss' Kappa = 0.43 (0.27, 0.59)). The results of McNemar's test, using a 40 mg/L cut-off for a positive test, indicated that the frequency of obtaining a positive or negative result did not differ between the Afinion and Actim tests (p = 0.80), whereas it did differ with the Afinion and BTNX tests (p < 0.0001).

The Duo CRP RDT test was also evaluated using Afinion as the reference test. At its cut-off of 20 mg/L, the RDT demonstrated moderate to strong agreement with the Afinion test (simple Kappa = 0.71 (0.51, 0.92)). The sensitivity of the test was 68.4% (48.5%, 89.3%), and the specificity was 100%.

## Malaria and CRP

Malaria prevalence, measured by the mRDT used for routine clinical care, was 29% (43/150). Concordance between that mRDT and the Duo malaria RDT was high, with only 2% (1/50) of tests classified differently (simple Kappa = 0.95 (0.84, 1)). Participants who tested positive for malaria by mRDT on average had higher CRP levels than those who tested negative (Fig 3). The mean CRP (SD) as measured using the Afinion assay in malaria positive and malaria negative patients was 20.5 (2.8) mg/L and 58.9 (8.8) mg/L, respectively.

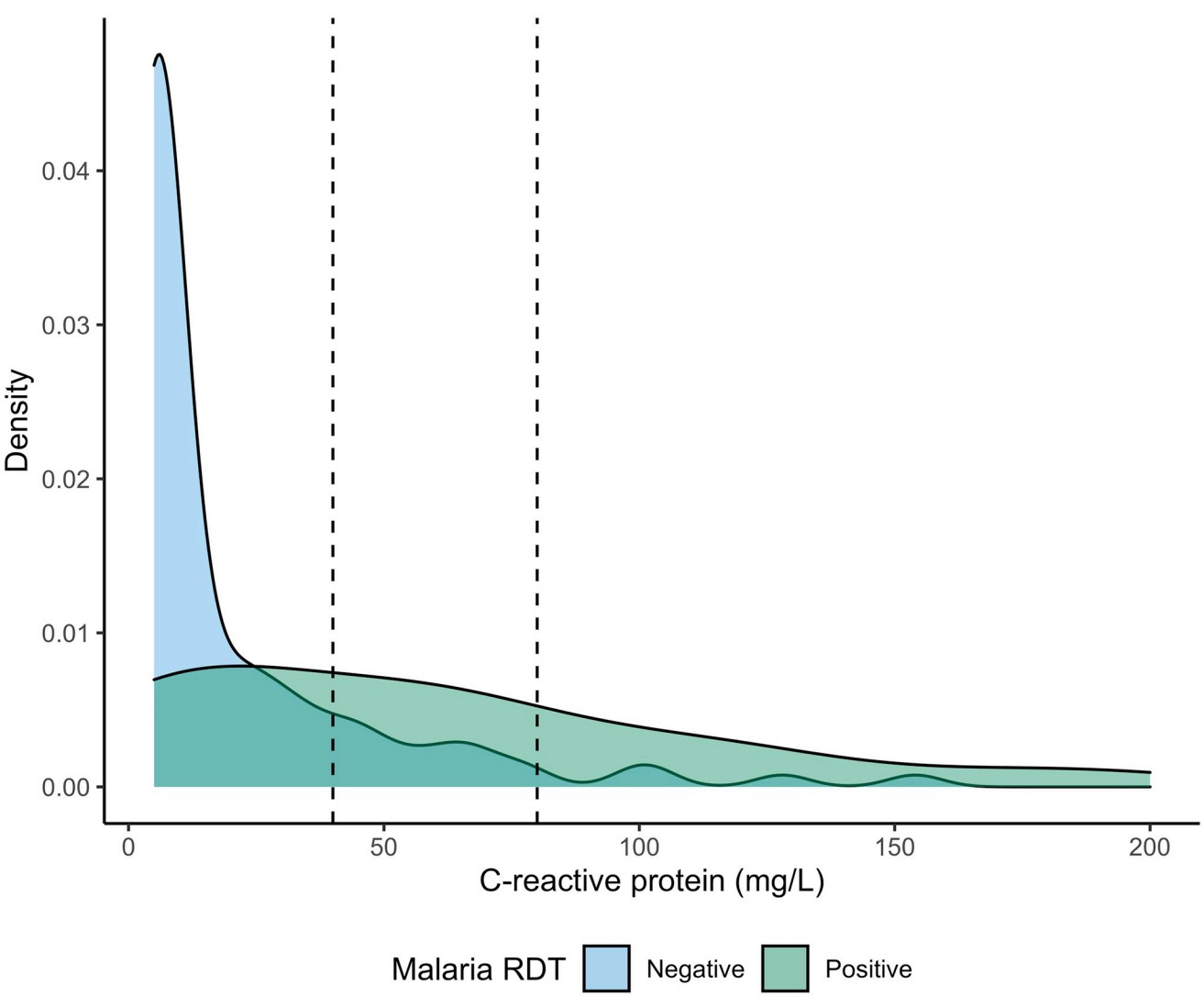

**Fig 3. Distribution of CRP by malaria status.** CRP measured by Afinion assay and malaria status measured by RDT in 134 study participants. Dashed lines indicate cut-offs of 40 mg/L and 80 mg/L.

## Discussion

The use of CRP RDTs in resource-limited settings has the potential to reduce inappropriate antibiotic use in children with febrile illness. However, it is imperative that these tests are rigorously assessed prior to their use in research studies or for clinical care to ensure data and care quality. In this study, we compared three CRP tests that can be conducted without electricity outside of formal laboratory settings to a facility-based CRP test requiring an electrically powered analyzer that served as the reference standard. Overall, the three RDTs demonstrated moderately strong agreement with the laboratory-based CRP test and performed well in classifying patients into CRP categories. For both use case scenarios (cut-offs of 40 and 80mg/L), the Actim test demonstrated a higher PPV as compared to the BTNX, whereas both tests had high NPVs (>90%). The Actim had fewer false positives at both cut-offs than the BTNX, whereas there were fewer false negatives with the BTNX, although confidence intervals overlapped for both comparisons. The concordance between the Actim and BTNX tests varied with each

other, as indicated by the Fleiss' Kappa statistic. The Duo CRP test perfectly categorized those with CRP <20 mg/L, though we tested fewer samples with Duo (n = 50) than with Actim and BTNX (n = 150) and mismeasurement may have occurred in the larger samples with greater variability.

Our findings are congruent with the limited existing literature on semi-quantitative point-of-care CRP tests [24, 25]. Specifically, the agreement between Actim and the reference CRP test was similar to [24] and slightly higher [30] than the estimates reported in previous studies. Of note, the latter study reported simple instead of weighted kappa estimates. Further, similarly to a previous study, we found that the agreement between both the Actim and BTNX semi-quantitative tests and the reference test was higher in the lower and upper categories (<10 and >80 mg/L), compared to the middle categories (10–40 and 40–80 mg/L), supporting the conclusion the "best" test may vary based on the use case and CRP cut-off needed [24]. In other words, the clinical context to which the test is being applied influences whether maximizing sensitivity or specificity should be prioritized, which in turn informs the choice of the most appropriate CRP cut-off. The assay with the best agreement at that chosen cut-off would then be the preferred test. In addition, the use of CRP RDTs may be best implemented in a multi-stage clinical algorithm. To minimize false negatives and limit adverse outcomes, a highly sensitive initial test should be performed first. Then, a highly specific CRP RDT may be used to distinguish whether a patient needs antibiotics to minimize false positives and reduce antibiotic prescription. Previous studies have demonstrated the use of such algorithms [21, 23].

A key strength of this study is the comparison of commercially available CRP RDTs in the laboratory of a rural, peripheral health facility in a resource-constrained setting and, in particular, in SSA. Our design was similar to a previous study comparing CRP RDTs that was conducted in Laos but evaluated different RDTs [25]. Their analysis found similar weighted kappa values to what we observed for the Actim and BTNX RDTs. Our data speak to the performance of these tests in routine practice, extending the work of other studies that assessed analytic performance of the same CRP RDTs (among others), but under more controlled conditions [24, 30].

Although not the primary focus of our study, we noted an important finding related to CRP measurement in this malaria-endemic region. The mean quantitative CRP was over twice as high in mRDT-positive as compared to mRDT-negative individuals. This trend has been seen in previous studies as well; CRP is produced in response to malaria infection and tends to be elevated in patients with parasitemia [44, 45]. As CRP levels in malaria-infected individuals can be similar to those seen in the context of bacterial infection, the presence of parasitemia should be taken into consideration when applying CRP to differentiate bacterial from viral infections [17, 46]. Further research is needed to determine the appropriate CRP threshold for antibiotic treatment in the context of malaria infection [27].

There are limitations to our study. First, missing Afinion test results for 11% of children enrolled due to power availability issues reduced the number of paired samples available for analysis. This further highlights the need for RDTs that do not require electricity for use in resource-limited contexts like rural peripheral health centers. Next, an Afinion analyzer was used as the reference test, which is considered a POC test as well as it can be used in clinical settings outside of a formal clinical laboratory, such as in primary care. Previous studies, however, have demonstrated the high accuracy of the Afinion in measuring CRP in children as compared to conventional methodologies [30–32]. Due to the availability of study personnel, the testing was performed by one of three laboratory technicians, though one technician performed all tests for each patient. A previous study of the Actim CRP tests noted interobserver variability in interpretation of the test, so this could have affected our results [30]. Further, the

time of year may have affected CRP levels in the population, as well as the prevalence of elevated CRP levels. The relative seasonal prevalence of infectious causes of fever, which could affect concordance between tests, particularly at the boundaries of the categories of the semi-quantitative tests.

In summary, our study provides important additional information about the performance of three rapid, point-of-care CRP tests in a resource-constrained setting that can inform selection of the appropriate assay for future research and clinical use cases. Overall, CRP RDTs demonstrate substantial agreement and appear to be promising tools for improving antibiotic provision in the context of pediatric febrile illness, warranting further assessment of their cost-effectiveness and impact on clinical outcomes.

## Supporting information

**S1 Text. Case report form completed for all enrolled participants.**
(PDF)

**S1 Checklist. Completed inclusivity in global health research questionnaire.**
(DOCX)

## Acknowledgments

We would like to thank the children and their caregivers for participating in this study and the clinical and laboratory staff at Bugoye Health Centre III for their assistance with implementation of the study. In addition, we would like to thank Medilink Lab & Surgicals Limited (Kampala, Uganda) for the donation of the Duo test kits.

## Author Contributions

**Conceptualization:** Jonathan J. Juliano, Edgar M. Mulogo, Ross M. Boyce, Emily J. Ciccone.

**Data curation:** Bradley Lin, Emily J. Ciccone.

**Formal analysis:** Caitlin A. Cassidy, Bradley Lin, Brandon Hollingsworth.

**Funding acquisition:** Jonathan J. Juliano, Edgar M. Mulogo, Emily J. Ciccone.

**Investigation:** Lydiah Kabugho, Georget Kibaba, Emily J. Ciccone.

**Methodology:** Caitlin A. Cassidy, Brandon Hollingsworth, Jonathan J. Juliano, Ross M. Boyce, Emily J. Ciccone.

**Project administration:** Lydiah Kabugho, Georget Kibaba, Emmanuel Baguma, Emily J. Ciccone.

**Resources:** Emily J. Ciccone.

**Software:** Caitlin A. Cassidy.

**Supervision:** Emmanuel Baguma, Jonathan J. Juliano, Edgar M. Mulogo, Ross M. Boyce, Emily J. Ciccone.

**Visualization:** Caitlin A. Cassidy.

**Writing – original draft:** Caitlin A. Cassidy, Emily J. Ciccone.

**Writing – review & editing:** Caitlin A. Cassidy, Lydiah Kabugho, Georget Kibaba, Brandon Hollingsworth, Jonathan J. Juliano, Edgar M. Mulogo, Ross M. Boyce, Emily J. Ciccone.

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
