## [Decision Letter · Decision Letter 0]

14 Sep 2023

PGPH-D-23-00899

Comparison of commercially available, rapid, point-of-care C-reactive protein assays among children with febrile illness in southwestern Uganda

Dear Dr. Ciccone,

Thank you for submitting your manuscript to PLOS Global Public Health. After careful consideration, we feel that it has merit but does not fully meet PLOS Global Public Health’s publication criteria as it currently stands. Therefore, we invite you to submit a revised version of the manuscript that addresses the points raised during the review process.

We look forward to receiving your revised manuscript.

Kind regards,

Jianhong Zhou

Staff Editor

Journal Requirements:

2. We noticed that you used "data not shown" in the manuscript. We do not allow these references, as the PLOS data access policy requires that all data be either published with the manuscript or made available in a publicly accessible database. Please amend the supplementary material to include the referenced data or remove the references.

3. We have noticed that you have a list of Supporting Information legends in your manuscript. However, there are no corresponding files uploaded to the submission. Please upload them as separate files with the item type 'Supporting Information'.

Additional Editor Comments (if provided):

Reviewers' comments:

Reviewer's Responses to Questions

**Comments to the Author**

1. Does this manuscript meet PLOS Global Public Health’s publication criteria? Is the manuscript technically sound, and do the data support the conclusions? The manuscript must describe methodologically and ethically rigorous research with conclusions that are appropriately drawn based on the data presented.

Reviewer #1: Partly

Reviewer #2: Yes

2. Has the statistical analysis been performed appropriately and rigorously?

Reviewer #1: Yes

Reviewer #2: I don't know

3. Have the authors made all data underlying the findings in their manuscript fully available (please refer to the Data Availability Statement at the start of the manuscript PDF file)?

Reviewer #1: No

Reviewer #2: Yes

4. Is the manuscript presented in an intelligible fashion and written in standard English?

Reviewer #1: Yes

Reviewer #2: Yes

5. Review Comments to the Author

Reviewer #1: This manuscript describes a study that evaluated the analytical performance of 3 RDT tests for CRP. The evaluation was performed using samples from 150 children presenting with febrile symptoms. The authors reported that the tests showed acceptable performance compared with the reference test.

The study is well designed and clear, but it would be useful if the authors could provide more information or explanation as follows:

-The author does not explain why these three specific tests were selected rather than others. It would be helpful to understand the reasons behind this choice.

- Could the author provide more details about the testing procedure? Were all analyses (RDTs and reference standards) performed by the same laboratory technician? Was the technician blinded to the results of Afinion (reference standard)? If not, this could be a limitation of the study and should be mentioned in the discussion.

-The author could discuss better the complexity of having a "universal" threshold . The author should explain why he/she decided to analyze 40 and 80 as cut-offs (why not include 20 to have a better comparison between the 3 RDTs)?

- Are N=50 samples sufficient to make statements about the performance of the Duo RDT?

- The caption of Figure 1 (line 190) should include more details about the analysis performed.

-Row 186, typo "performance".

Reviewer #2: In this paper, the authors compare three semi-quantitative CRP RDTs to Afinion as goal standard. With the inappropriate use of antibiotics in LMICs due to the lack of appropriate and practical diagnostic tool to discriminate bacterial infections to other infections, CRP can be an alternative to support this diagnostic approach. The semi-quantitative CRP is easy to use by non-technician and can be perform outside conventional lab. Nevertheless, there is a need to assess the performance of these semi-quantitative CRP. I found this paper interesting and it is a great pleasure to read this paper.

Just one point that the authors can mentionne is the production of CRP in malaria infected people. This can make the implementation difficult in malaria endemic area, if an approach with positivity cut-offs are not determined for bacterial infections in these areas.

Except that point, I think this paper can be published.

6. PLOS authors have the option to publish the peer review history of their article (what does this mean?). If published, this will include your full peer review and any attached files.

**Do you want your identity to be public for this peer review?** For information about this choice, including consent withdrawal, please see our Privacy Policy.

Reviewer #1: No

Reviewer #2: No

---

## [Editor Report · Decision Letter 1]

28 Nov 2023

Comparison of commercially available, rapid, point-of-care C-reactive protein assays among children with febrile illness in southwestern Uganda

PGPH-D-23-00899R1

Dear Dr. Ciccone,

We are pleased to inform you that your manuscript 'Comparison of commercially available, rapid, point-of-care C-reactive protein assays among children with febrile illness in southwestern Uganda' has been provisionally accepted for publication in PLOS Global Public Health.

Best regards,

Nei-yuan (Marvin) Hsiao

Academic Editor